# SEMI-SUPERVISED LEARNING WITH GANS: REVISITING MANIFOLD REGULARIZATION

**Bruno Lecouat**[*,1,2]**Chuan-Sheng Foo**[*,2]**, Houssam Zenati**[2,3]**, Vijay R. Chandrasekhar**[2,4]

[1] Télécom ParisTech, `bruno.lecouat@gmail.com`.

[2] Institute for Infocomm Research, Singapore, {`foocs, vijay`}`@i2r.a-star.edu.sg`.

[3] CentraleSupélec, `houssam.zenati@student.ecp.fr`.

[4] School of Computer Science, Nanyang Technological University.

[*] Equal contribution

## ABSTRACT

GANS are powerful generative models that are able to model the manifold of natural images. We leverage this property to perform manifold regularization by approximating the Laplacian norm using a Monte Carlo approximation that is easily computed with the GAN. When incorporated into the feature-matching GAN of Salimans et al. (2016), we achieve state-of-the-art results for GAN-based semi-supervised learning on the CIFAR-10 dataset, with a method that is significantly easier to implement than competing methods.

## 1 INTRODUCTION

Generative adversarial networks (GANs) are a powerful class of deep generative models that are able to model distributions over natural images. The ability of GANs to generate realistic images from a set of low-dimensional latent representations, and moreover, plausibly interpolate between points in the low-dimensional space (Radford et al., 2016; J.-Y. Zhu & Efros, 2016), suggests that they are able to learn the manifold over natural images.

In addition to their ability to model natural images, GANs have been successfully adapted for semi-supervised learning, typically by extending the discriminator to also determine the specific class of an example (or that it is a generated example) instead of only determining whether it is real or generated (Salimans et al., 2016). GAN-based semi-supervised learning methods have achieved state-of-the-art results on several benchmark image datasets (Dai et al., 2017; Li et al., 2017).

In this work, we leverage the ability of GANs to model the manifold of natural images to efficiently perform manifold regularization through a Monte-Carlo approximation of the Laplacian norm (Belkin et al., 2006). This regularization encourages classifier invariance to local perturbations on the image manifold as parametrized by the GAN's generator, which results in nearby points on the manifold being assigned similar labels. We applied this regularization to the semi-supervised feature-matching GAN of Salimans et al. (2016) and achieved state-of-the-art performance amongst GAN-based methods on the SVHN and CIFAR-10 benchmarks.

## 2 RELATED WORK

Belkin et al. (2006) introduced the idea of manifold regularization, and proposed the use of the Laplacian norm $\|f\|_L^2 = \int_{x \in M} \|\nabla_{\mathcal{M}} f(x)\|^2 \, d\mathcal{P}_X(x)$ to encourage local invariance and hence smoothness of a classifier $f$, at points on the data manifold $\mathcal{M}$ where data are dense (i.e., where the marginal density of data $P_X$ is high). They also proposed graph-based methods to estimate the norm and showed how it may be efficiently incorporated with kernel methods.

In the neural network community, the idea of encouraging local invariances dates back to the TangentProp algorithm of Simard et al. (1998), where manifold gradients at input data points are es-

---

[*]All code and hyperparameters may be found at `https://github.com/bruno-31/GAN-manifold-regularization`

timated using explicit transformations of the data that keep it on the manifold (like small rotations and translations). Since then, contractive autoencoders (Rifai et al., 2011) and most recently GANs (Kumar et al., 2017) have been used to estimate these gradients directly from the data. Kumar et al. (2017) also add an additional regularization term which promotes invariance of the discriminator to *all* directions in the data space; their method is highly competitive with the state-of-the-art GAN method of Dai et al. (2017).

## 3    METHOD

The key challenge in applying manifold regularization is in estimating the Laplacian norm. Here, we present an approach based on the following two empirically supported assumptions: 1) GANs can model the distribution of images (Radford et al., 2016), and 2) GANs can model the image manifold (Radford et al., 2016; J.-Y. Zhu & Efros, 2016). Suppose we have a GAN that has been trained on a large collection of (unlabeled) images. Assumption 1 implies that a GAN approximates the marginal distribution $\mathcal{P}_X$ over images, enabling us to estimate the Laplacian norm over a classifier $f$ using Monte Carlo integration with samples $z^{(i)}$ drawn from the space of latent representations of the GAN's generator $g$. Assumption 2 then implies that the generator defines a manifold over image space (Shao et al., 2017), allowing us to compute the required gradient (Jacobian matrix) with respect to the latent representations instead of having to compute tangent directions explicitly as in other methods. Formally, we have

$$\|f\|_L^2 = \int_{x \in M} \|\nabla_{\mathcal{M}} f(x)\|^2 \, d\mathcal{P}_X(x) \overset{(1)}{\approx} \frac{1}{n} \sum_{i=1}^{n} \left\| \nabla_{\mathcal{M}} f(g(z^{(i)})) \right\|^2 \overset{(2)}{\approx} \frac{1}{n} \sum_{i=1}^{n} \left\| J_z f(g(z^{(i)})) \right\|_F^2$$

where the relevant assumption is indicated above each approximation step, and $J$ is the Jacobian.

In our experiments, we used stochastic finite differences to approximate the final Jacobian regularizer, as training a model with the exact Jacobian is computationally expensive. Concretely, we used $\frac{1}{n} \sum_{i=1}^{n} \left\| f(g(z^{(i)}) - f(g(z^{(i)} + \epsilon \bar{\delta})) \right\|_F^2$, where $\bar{\delta} = \frac{\delta}{\|\delta\|}$, $\delta \sim \mathcal{N}(0, I)$. We tuned $\epsilon$ using a validation set, and the final values used are reported in the Appendix (Table 4).

Unlike the other methods discussed in the related work, we do not explicitly regularize at the input data points, which greatly simplifies its implementation. For instance, in Kumar et al. (2017) regularizing on input data points required the estimation of the latent representations for each input as well as several other tricks to workaround otherwise computationally expensive operations involved in estimating the manifold tangent directions.

We applied our Jacobian regularizer to the discriminator of a feature-matching GAN adapted for semi-supervised learning (Salimans et al., 2016); the full loss being optimized is provided in the Appendix. We note that this approach introduces an interaction between the regularizer and the generator with unclear effects, and somewhat violates assumption 1 since the generator does not approximate the data distribution well at the start of the training process. Nonetheless, the regularization provided improved classification accuracy in our experiments and the learned generator does not seem to obviously violate our assumptions as demonstrated in the Appendix (Figure 1). We leave a more thorough investigation for future work [1].

## 4    EXPERIMENTS

We evaluated our method on the SVHN (with 1000 labeled examples) and CIFAR-10 (with 4000 labeled examples) datasets. We used the same network architecture as Salimans et al. (2016) but we tuned several training parameters: we decreased the batch size to 25 and increased the number of maximum training epochs. Hyperparameters were tuned using a validation set split out from the training set, and then used to train a final model on the full training set that was then evaluated. A detailed description of the experimental setup is provided in the Appendix. We report error rates on the test set in Table 1.

---

[1]One alternative approach that avoids this interaction would be to first train a GAN to convergence and use it to compute the regularizer while training a separate classifier network.

Table 1: Comparison of error rate with state-of-the-art methods on two benchmark datasets. $N_l$ indicates the number of labeled examples in the training set. Only results *without* data augmentation are included. Our reported results are from 5 runs with different random seeds.

| Method | SVHN (%) $N_l$=1000 | CIFAR-10 (%) $N_l$=4000 |
|---|---|---|
| Ladder network, Rasmus et al. (2015) | | $20.40 \pm 0.47$ |
| Π model, Laine & Aila (2017) | $5.43 \pm 0.25$ | $16.55 \pm 0.29$ |
| VAT (large) Miyato et al. (2017) | $5.77$ | $14.82$ |
| VAT+EntMin(Large), Miyato et al. (2017) | $4.28$ | $13.15$ |
| CatGAN, Springenberg (2016) | | $19.58 \pm 0.58$ |
| Improved GAN, Salimans et al. (2016) | $8.11 \pm 1.3$ | $18.63 \pm 2.32$ |
| Triple GAN, Li et al. (2017) | $5.77 \pm 0.17$ | $16.99 \pm 0.36$ |
| Improved semi-GAN, Kumar et al. (2017) | $4.39 \pm 1.5$ | $16.20 \pm 1.6$ |
| Bad GAN, Dai et al. (2017) | $4.25 \pm 0.03$ | $14.41 \pm 0.30$ |
| Improved GAN (ours) | $5.6 \pm 0.10$ | $15.5 \pm 0.35$ |
| **Improved GAN (ours) + Manifold Reg** | $\mathbf{4.51 \pm 0.22}$ | $\mathbf{14.45 \pm 0.21}$ |

Surprisingly, we observe that our implementation of the feature-matching GAN ("Improved GAN") significantly outperforms the original, highlighting the fact that GAN training is sensitive to training hyperparameters. Incorporating our Jacobian manifold regularizer further improves performance, leading our model to achieve state-of-the-art performance on CIFAR-10, as well as being extremely competitive on SVHN. In addition, our method is dramatically simpler to implement than the state-of-the-art GAN method BadGAN (Dai et al., 2017) that requires the training of a PixelCNN.

## 5 CONCLUSION

We leveraged the ability of GANs to model natural image manifolds to devise a simple and fast approach to manifold regularization by Monte-Carlo approximation of the Laplacian norm. When applied to the feature-matching GAN, we achieve state-of-the-art performance amongst GAN-based methods for semi-supervised learning. Our approach has the key advantage of being simple to implement, and scales to large amounts of unlabeled data, unlike graph-based approaches.

We plan to study the interaction between our Jacobian regularization and the generator; as even though the loss back-propagates only to the discriminator, it indirectly affects the generator through the adversarial training process. Determining whether our approach is effective for semi-supervised learning in general, by using a GAN to regularize a separate classifier is another interesting direction for future work.

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

APPENDIX

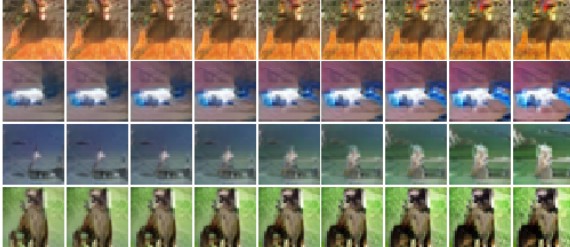

Figure 1: Linear interpolation between two random directions in the latent representation space of a generator trained with our Jacobian regularization.

OBJECTIVE FUNCTION OF THE GAN

We use the following loss function for the discriminator:

$$L = L_{supervised} + L_{unsupervised}$$

$$L_{supervised} = -\mathbb{E}_{x,y \sim p_{data}(x,y)} \left[ \log\ p_f(y|x, y < K+1) \right]$$
$$L_{unsupervised} = -\mathbb{E}_{x \sim p_{data}(x)} \left[ \log\ \left[ 1 - p_f(y = K+1|x) \right] \right] - \mathbb{E}_{x \sim g} \left[ \log\ \left[ p_f(y = K+1|x) \right] \right]$$
$$+ \lambda\ \mathbb{E}_{z \sim U(z), \delta \sim N(\delta))} \left\| f(g(z)) - f(g(z + \epsilon\bar{\delta})) \right\|_2^2$$

For the generator we use the feature matching loss (Salimans et al., 2016): $\left\| \mathbb{E}_{x \sim p_{data}} h(x) - \mathbb{E}_{z \sim p_z(z)} h(g(z)) \right\|_1$. Here, $h(x)$ denotes activations on an intermediate layer of the discriminator. In our experiments, the activation layer after the Networks in Networks (NiN) layers (Lin et al., 2014) was chosen as intermediate $h(x)$.

SEMI-SUPERVISED CLASSIFICATION FOR CIFAR-10 AND SVHN DATASET

The CIFAR-10 dataset (Krizhevsky, 2009) consists of 32*32*3 pixel RGB images of objects categorized by their corresponding labels. The number of training and test examples in the dataset are respectively 50,000 and 10,000.

The SVHN dataset (Yuval Netzer, 2011) consists of 32*32*3 pixel RGB images of numbers categorized by their labels. The number of training and test examples in the dataset are 73,257 and 26,032 respectively.

To do fair comparisons with other methods we did not apply any data augmentation on training data. We randomly picked a small fraction of images from the training set as labeled examples and kept another fraction of it as the validation set. Then we used early stopping on the validation set. Finally, we trained on the merged training set and validation set with different random seed and stopped the training on the previously found number of epochs (with the validation set) and report the results on the testing set (see Table 4)

NEURAL NET ARCHITECTURE

We used the same architecture as the one proposed in Salimans et al. (2016). For the discriminator in our GAN we used a 9 layer deep convolutional network with dropout (Srivastava et al., 2014) and weight normalization (Salimans & Kingma, 2016). The generator was a 4 layer deep convolutional neural network with batch normalization (Ioffe & Szegedy, 2015). We used an exponential moving average of the parameters for the inference on the testing set. The architecture of our model is described in table 2 & 3.
We only made minor changes to the training procedure – we reduced batch size (100 to 25 and 50)

and slightly increased the number of training epochs. The learning rate is linearly decayed to 0. Finally, we initialized the weights of the discriminator weight norm layer in a different way. In their paper, the authors initialized their weights in a data-driven way. Instead, we set the neuron bias $b$ to 0 rather than $\frac{-\mu[t]}{\sigma[t]}$ and the scale $g$ is set to 1 rather than $\frac{1}{\sigma[t]}$. For our regularization term, we use the same number of samples $n$ for the Monte Carlo estimator as the batch size in the stochastic gradient descent.

Table 2: Discriminator architecture

| conv-large CIFAR-10 | conv-small SVHN |
|---|---|
| 32×32×3 RGB images | |
| dropout, $p = 0.2$ | |
| 3×3 conv. weightnorm 96 lReLU | 3×3 conv. weightnorm 64 lReLU |
| 3×3 conv. weightnorm 96 lReLU | 3×3 conv. weightnorm 64 lReLU |
| 3×3 conv. weightnorm 96 lReLU stride=2 | 3×3 conv. weightnorm 64 lReLU stride=2 |
| dropout, $p = 0.5$ | |
| 3×3 conv. weightnorm 192 lReLU | 3×3 conv. weightnorm 128 lReLU |
| 3×3 conv. weightnorm 192 lReLU | 3×3 conv. weightnorm 128 lReLU |
| 3×3 conv. weightnorm 192 lReLU stride=2 | 3×3 conv. weightnorm 128 lReLU stride=2 |
| dropout, $p = 0.5$ | |
| 3×3 conv. weightnorm 192 lReLU pad=0 | 3×3 conv. weightnorm 128 lReLU pad=0 |
| NiN weightnorm 192 lReLU | NiN weightnorm 128 lReLU |
| NiN weightnorm 192 lReLU | NiN weightnorm 128 lReLU |
| global-pool | |
| dense weightnorm 10 | |

Table 3: Generator architecture

| CIFAR-10 & SVHN |
|---|
| latent space 100 (uniform noise) |
| dense $4 \times 4 \times 512$ batchnorm ReLU |
| 5×5 conv.T stride=2 256 batchnorm ReLU |
| 5×5 conv.T stride=2 128 batchnorm ReLU |
| 5×5 conv.T stride=2 3 weightnorm tanh |

Table 4: Hyperparameters determined on validation set

| Hyper-parameter | SVHN | CIFAR-10 |
|---|---|---|
| $\lambda$ regularization weight | $10^{-3}$ | $10^{-3}$ |
| $\epsilon$ norm of perturbation | $10^{-5}$ | $10^{-5}$ |
| Epoch (early stopping) | 400 (312) | 1400 (1207) |
| Batch size | 50 | 25 |
| Monte Carlo sampling size | 50 | 25 |
| Leaky ReLU slope | 0.2 | 0.2 |
| Learning rate decay | linear decay to 0 after 300 epochs | linear decay to 0 after 1200 epochs |
| Optimizer | ADAM ($\alpha = 3 * 10^{-4}, \beta_1 = 0.5$) | |
| Weight initialization | Isotropic gaussian ($\mu = 0, \sigma = 0.05$) | |
| Bias initialization | Constant (0) | |

