# OpenReview forum: "Semi-Supervised Learning With GANs: Revisiting Manifold Regularization"
_ICLR.cc/2018/Workshop — Accept_

### Official Review · AnonReviewer2 · 2018-03-07
**interesting synthesis of prior methods**

**Rating:** 8
**Confidence:** 4

**Review:**

I think this is a neat implementation of an intuitively-appealing idea -- i.e., we can regularize a classifier by encouraging its predictions to change slowly along trajectories that follow the data manifold.

The general idea of optimizing monte-carlo approximations of a classifier's sensitivity to perturbations -- in its inputs, parameters, architecture, etc -- has been explored in previous works, e.g. "Learning with Pseudo Ensembles" (Bachman et al., NIPS 2014), "Temporal Ensembling..." (Laine et al., ICLR 2017), or "Mean Teachers..." (Tarvainen et al., NIPS 2017).

The trick in this paper is to construct a set of perturbations that approximate small displacements along the data manifold. The authors do this by generating an image using an adversarially-trained generator and perturbing its "latent code". The classifier is trained to minimize the sensitivity of its predictions to these perturbations. The authors add this regularization on top of the method from Salimans et al. (NIPS 2016). This produces strong results on two standard semi-supervised learning benchmarks.

One challenge I see for this approach is that it may depend strongly on being able to generate sufficiently good images. If the authors could design an experiment to measure how the proposed regularization's performance degrades as the generator becomes worse, that would be cool.

---

### Official Review · AnonReviewer1 · 2018-03-08
**Interesting paper**

**Rating:** 8
**Confidence:** 4

**Review:**

The paper proposes to use the manifold regularization to further improve current SSL GAN models. Abilities of GANs to model  image manifolds provide an efficient way to approximate the Laplacian norm. The improvement over the baseline model is significant and the final result is comparable to the state-of-the-art. I'm also interested in the interaction among the regularization, generator and classifier in the feature-matching GAN framework, which is a very nice follow-up.

Empirically, it is better to show that the regularization can consistently improve the SSL results with a broader family of algorithms, architectures and hyperparameters.

Anyway, the idea is very interesting and it should be accepted.

---

### Official Review · AnonReviewer3 · 2018-03-10
**Interesting idea tying together GANs and classic manifold learning work. Some baselines are missing.**

**Rating:** 7
**Confidence:** 3

**Review:**

The paper uses widely held assumptions about GANs in an interesting way: the assumption that GANs capture the underlying manifold structure of natural images embedded in Euclidean space, as well as the distribution of natural images over this manifold. While this assumption has been contested (Arora, Ge, Liang, Ma & Jang 2017, Arora & Zhang 2017), it seems reasonable to see if it one can make progress based on making this assumption.

The way the paper uses this assumption is using the idea of manifold regularization popularized by Belkin et al. (2016), noting that GANs offer a potentially more powerful way to model the Laplacian compared with the classic, spectral methods.

Experiments seem to indicate this idea has merit, showing strong performance. Somewhat disconcerting is the strong dependence of some of the baselines on hyperparameter selection - this is not a bug of the current paper, rather a bug of the field of GAN experiments, and I thank the authors for reporting this additional finding.

An important missing baseline is comparing to classifiers using spectral Laplacian methods. One could easily train a standard DNN on a classification task and calculate the Laplacian over on the layers, using it along the lines of Belkin et al.

Pros:
1. Interesting idea, novel to the best of my knowledge.
2. Good experimental results.

Cons:
1. Missing baseline
2. Lacking discussion of the approximation method for the Jacobian. Is the Gaussian sampling acting simply as some sort of regularizer, or is the Jacobian structure truly used?  How sensitive are the results to the choice of epsilon? What would happen if instead of the "Jacobian-like" term one would use a simply the norm ||f|| implied by the ambient Euclidean space?

---

### Decision · Program_Chairs · 2018-03-20
**ICLR 2018 Workshop Acceptance Decision**

**Decision:**

Accept

**Comment:**

Congratulations, your paper was accepted to the ICLR workshop.